# DOMAIN FEATURE PERTURBATION FOR DOMAIN GENERALIZATION

## ABSTRACT

Deep neural networks (DNNs) often struggle with distribution shifts between training and test environments, which can lead to poor performance, untrustworthy predictions, or unexpected behaviors. In this work, we propose domain feature perturbation (DFP), a novel approach that explicitly leverages domain information to improve the out-of-distribution performance of DNNs. Specifically, we train a domain classifier in conjunction with the main prediction model and perturb the multi-layer representation of the latter with random noise modulated by the gradient of the former. The domain classifier is designed to share the backbone with the main model and is easy to implement with minimal extra model parameters that can be discarded at inference time. Intuitively, our proposed method aims to reduce the dependence of the main prediction model on domain-specific features, such that the model can focus on domain-agnostic features that generalize across different domains. We demonstrate the effectiveness of DFP on multiple benchmarks for domain generalization.

## 1 INTRODUCTION

Deep neural networks (DNNs) have exhibited impressive performance in solving a wide variety of real-world tasks. A crucial aspect of the success lies in their ability to learn from a large amount of training data (Shinde & Shah, 2018). However, despite the widely held assumption that training and test data are sampled from the same underlying probability distribution (Golden, 2020), real-world data frequently exhibits deviations from the training data distribution. Consequently, a shift occurs in the distribution between the training and test data. Such shifts are prevalent in diverse tasks, examples include face recognition under varying lighting conditions or backgrounds (Adjabi et al., 2020), medical image recognition with different imaging devices or acquisition protocols (Zhou et al., 2022), and autonomous driving in different cities (Sun et al., 2020). In these cases, the presence of out-of-distribution (OOD) data poses significant challenges to conventional supervised learning methods, leading to inaccurate and unreliable predictions.

To address the challenges of OOD generalization, researchers have explored various techniques, such as meta-learning (Bui et al., 2021), causal learning (Mahajan et al., 2021; Lv et al., 2022), contrastive learning (Kim et al., 2021), and disentangled representation learning (Zhang et al., 2022). In particular, when given access to training data that is split into multiple domains and expected to generalize to an unseen test domain, which is known as *domain generalization*, one can take advantage of domain information (i.e. domain labels) to achieve better OOD performance. However, existing approaches to domain generalization have shown limited success in utilizing such information (Gulrajani & Lopez-Paz, 2020; Ye et al., 2022), or rely on complex training procedures that involve adversarial training (Ganin et al., 2016; Lee et al., 2019; Liu et al., 2021) or separate auxiliary models (Liu et al., 2021; Bui et al., 2021; Zhang et al., 2022).

In this work, we propose a novel approach to leveraging domain information. Specifically, we train a domain classifier in conjunction with the main prediction model and perturb the multi-layer representation of the latter with noise modulated by the gradient of the former. The domain classifier is designed to share the backbone with the main model, introducing minimal extra model parameters that can be discarded at inference time. This technique aims to reduce the dependence of the main prediction model on domain-specific features, such that the model can focus on domain-agnostic features that generalize across different domains. The resulting method, named *Domain Feature*

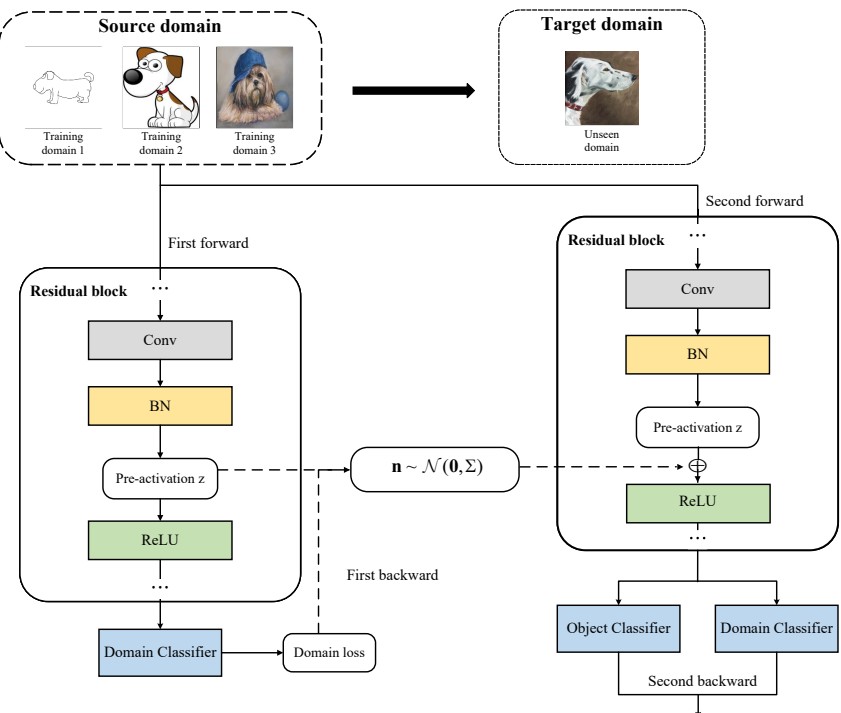

Figure 1: Overview of the training procedure of DFP. The two classification heads share a single backbone network. The first backward pass generates the modulated perturbation, followed by a second backward pass that updates the model parameters.

*Perturbation (DFP)*, is easy to implement without major modifications to the model architecture and shows significantly improved performance on multiple domain generalization benchmarks. We summarize our main contributions as follows:

- We explicitly leverage domain information by training a domain classifier along with the main model. We then compute the gradient of the domain classifier with respect to intermediate representations, the magnitude of which is utilized to identify domain-specific features in a relatively simple manner.

- We propose to perturb intermediate representations with random noise modulated by the aforementioned gradient magnitude. By doing so, domain-specific features are automatically identified and randomized, effectively reducing the dependence of the main model on these features. In addition, we propose a gradient similarity measure to evaluate such regularization effect.

- We evaluate our method on multiple domain generalization benchmarks following a recently proposed evaluation protocol (Ye et al., 2022) for a fair comparison with other methods. The experimental results demonstrate competitive or better performance compared to state-of-the-art methods.

## 2 METHODS

In this section, we first detail the basic concepts crucial to our approach, including out-of-distribution (OOD) generalization and diversity shift. Subsequently, we present our method in detail and define gradient similarity for the purpose of evaluation. Figure 1 illustrates the overall framework of our method.

## 2.1 PRELIMINARIES

**OOD generalization**  We begin by framing the general OOD generalization problem. Let $X$ denote the input space and $Y$ the target label space. We define a parametric model $f_\theta : X \to Y$, mapping input features to labels with parameters $\theta$. The loss function $L : \theta \times (X \times Y) \to \mathbb{R}$ quantifies the discrepancy between the predicted and true labels. Given a supervised learning task with $N$ training samples, $\{(x_i, y_i)\}_{i=1}^N$, where $x_i \in X$ and $y_i \in Y$, and sourced from the training distribution $P_{tr}(X, Y)$, the objective of OOD generalization is to identify a model that generalizes effectively to data from the test distribution $P_{te}(X, Y)$. However, without access to $P_{te}(X, Y)$ at training time, the model is usually optimized to minimize the empirical risk on $P_{tr}(X, Y)$:

$$\hat{f}_\theta = \arg\min_\theta \mathbb{E}_{X,Y \sim P_{tr}} L\left(f_\theta(X), Y\right). \tag{1}$$

In standard supervised learning, it's typically assumed that both training and test samples are i.i.d. samples from a shared distribution, denoted as $P_{tr}(X, Y) = P_{te}(X, Y)$. However, in the broader OOD context, training and test data may originate from different distributions, implying $P_{tr}(X, Y) \neq P_{te}(X, Y)$. For domain generalization tasks in particular, the combined training and test data are partitioned into $k$ domains, represented as $D = \{D_i\}_{i=1}^k$, with each domain stemming from a unique distribution. During training, $k - 1$ of these domains form the training set, while the remaining domain is held out for testing.

**Diversity shift**  Distribution shifts between training and test data can be categorized into diversify shift and correlation shift (Ye et al., 2022). Consider a training distribution $P_{tr}(X, Y)$ and a test distribution $P_{te}(X, Y)$ with probability functions $p$ and $q$, respectively. The labeling rule of the data, $f : X \to Y$, usually depends on a particular set of features $\mathcal{Z}_1$, whereas the rest of the features $\mathcal{Z}_2$ are not causal to the prediction of $Y$. That is, while both $\mathcal{Z}_1$ and $\mathcal{Z}_2$ jointly determines the input variable $X$, the target variable $Y$ is determined by $\mathcal{Z}_1$ alone. The following property for every $z \in \mathcal{Z}_1$ makes OOD generalization possible:

$$p(z) \cdot q(z) \neq 0 \land \forall y \in Y : p(y|z) = q(y|z). \tag{2}$$

And the opposite property of $z \in \mathcal{Z}_2$ makes OOD generalization challenging:

$$p(z) \cdot q(z) = 0 \lor \exists y \in Y : p(y|z) \neq q(y|z). \tag{3}$$

Diversity shift further assumes that $p(z) \cdot q(z) = 0$ for $z \in \mathcal{Z}_2$, meaning that the diversity of data is embodied by unique features not shared by the different domains. The extent of diversity shift can be measured by

$$D_{div}(p, q) := \frac{1}{2} \int_{\mathcal{S}} |p(z) - q(z)| \, \mathrm{d}z. \tag{4}$$

It is observed that many practical datasets for domain generalization exhibit significant diversity shifts (Ye et al., 2022). Therefore, exploiting the property of diversity shift may improve OOD generalization on such datasets.

## 2.2 DOMAIN FEATURE PERTURBATION

Given sufficient capacity, a parametric model, $f_\theta$, trained with the objective in Equation 1 is expected to capture the causal features in $\mathcal{Z}_1$, which is desirable for OOD generalization. However, $f_\theta$ may also rely on the non-causal features in $\mathcal{Z}_2$ if they correlate with $Y$. While the dependency on non-causal features can aid in i.i.d. generalization in the training environment, it is unlikely to generalize to the unseen test environment, especially considering the non-overlapping supports of $\mathcal{Z}_2$ caused by diversity shift (i.e. $p(z) \cdot q(z) = 0$ for $z \in \mathcal{Z}_2$). Theoretically, one can reduce the dependence on $\mathcal{Z}_2$ to achieve better OOD generalization. However, there is no guarantee that features from $\mathcal{Z}_1$ and $\mathcal{Z}_2$ captured by $f_\theta$ are well-separated, and identifying them is even more nontrivial. To tackle this challenge, we observe that, compared to $\mathcal{Z}_1$, the non-overlapping supports of $\mathcal{Z}_2$ across different domains make it particularly useful for domain classification. As such, we propose to

train a domain classifier to differentiate among different training domains, and use its gradient to approximately identify the features in $\mathcal{Z}_2$. Subsequently, we perturb the identified features with noise, reducing the dependence of $f_\theta$ on them in the training process. We refer to this approach as *domain feature perturbation (DFP)*.

Concretely, alongside the main classifier, we train a domain classifier to capture domain-specific features. The two classifiers can be implemented with two output heads that share the same backbone, and thus introduces negligible extra parameters. Let $g_\theta : X \rightarrow Y_d$ denote the domain classifier, where $Y_d$ is the set of training domain labels. The overall loss function is as follows:

$$\mathcal{L}(\theta) = \mathbb{E}_{X,Y,Y_d \sim P_{tr}} \left[ \alpha L\left(f_\theta(X), Y\right) + (1-\alpha)L_d\left(g_\theta(X), Y_d\right) \right], \tag{5}$$

where $L$ and $L_d$ correspond respectively to the main classifier and the domain classifier, and $\alpha \in (0, 1)$ is a hyperparameter to weight the two losses. To utilize the gradient of the domain classifier, the training process of DFP involves two forward passes and two backward passes in each iteration. In the first forward pass and backward pass, we compute $L_d$, as well as its gradient with respect to the intermediate representations of the backbone network, i.e.,

$$\nabla_{\mathbf{z}} L_d = \frac{\partial L_d}{\partial \mathbf{z}}, \tag{6}$$

where $\mathbf{z} \in \mathbb{R}^m$ represents the pre-activations of neurons. Let $z_i$ be the $i$-th element of $\mathbf{z}$, and $\epsilon$ a positive constant. The magnitude of $\nabla_{\mathbf{z}} L_d$ then serves to modulate a random noise vector $\mathbf{n} \sim \mathcal{N}(\mathbf{0}, \mathbf{\Sigma})$, such that

$$\mathbf{\Sigma} = \text{diag}\left(\sigma_1^2, \sigma_2^2, \ldots, \sigma_m^2\right), \text{and } \sigma_i = \frac{\epsilon}{\|\nabla_z L_d\|_p} \left| \frac{\partial L_d}{\partial z_i} \right|. \tag{7}$$

In the second forward pass, DFP applies modulated noise $\mathbf{n}$ to $\mathbf{z}$ at each layer as $\tilde{\mathbf{z}} = \mathbf{z} + \mathbf{n}$, and proceeds with the second backward pass to update model parameters. Note that in Equation 7, the $\sigma_i$'s are normalized by the $\ell_p$ norm of $\nabla_z L_d$ to keep the overall magnitude of $\mathbf{\Sigma}$ stable. In addition, $p$ is set to 2 throughout this paper unless otherwise specified. At test time, the domain classifier can be discarded, and the inference of the main classifier can be done in a single forward pass.

The intuition behind Equation 7 is that the features in $\mathcal{Z}_2$ are predominantly utilized by the domain classifier compared to those in $\mathcal{Z}_1$, and thus are more likely to have larger gradient magnitudes, $|\partial L_d/\partial z_i|$. By injecting noise with high variance to the features in $\mathcal{Z}_2$, we aim to diminish the reliance of $f_\theta$ on these features, thereby improving OOD generalization. The training procedure with DFP is detailed in Algorithm 1.

---

**Algorithm 1** Training procedure with domain feature perturbation (DFP)

**Input:** Main classifier $f_\theta$; Domain classifier $g_\theta$; Loss function $\mathcal{L}(\theta)$; Training data $D_{tr}$; Batch size $B$; Number of training epochs $E$.

**Training:**

1: **for** epoch $e = 1, 2, ..., E$ **do**
2:     **for** batch $b = b_1, b_2, ... \subset D_{tr}$ with batch size $B$ **do**
3:         $b_i = \{x_i, y_i, y_{d,i}\}$
4:         $\hat{y}_{d,i} = g_\theta(x_i)$                                        ▷ First forward
5:         $\nabla_{\mathbf{z}} L_d = \partial L_d(\hat{y}_{d,i}, y_{d,i})/\partial \mathbf{z}$                ▷ First backward
6:         $\mathbf{n} \sim \mathcal{N}(\mathbf{0}, \mathbf{\Sigma})$                    ▷ Sample DFP as per Equation 7
7:         $\hat{y}_i = f_\theta(x_i), \hat{y}_{d,i} = g_\theta(x_i)$ with $\tilde{\mathbf{z}} = \mathbf{z} + \mathbf{n}$     ▷ Second forward with DFP
8:         $\mathcal{L}(\theta) = \alpha L(\hat{y}_i, y_i) + (1-\alpha)L_d(\hat{y}_{d,i}, y_{d,i})$
9:         $\nabla_\theta \mathcal{L}(\theta) = \partial \mathcal{L}(\theta)/\partial \theta$                        ▷ Second backward
10:       $\theta \leftarrow \text{Optimizer}\left(\theta, \nabla_\theta \mathcal{L}(\theta)\right)$
11:     **end for**
12: **end for**

---

## 2.3 GRADIENT SIMILARITY

As discussed in Section 2.2, a model trained with DFP is expected to focus more on the features in $\mathcal{Z}_1$ than those in $\mathcal{Z}_2$. To examine if this is the case, one can compare the magnitudes of $L$'s gradients with respect to the two sets of features, which again requires separating and identifying the features in $\mathcal{Z}_1$ and $\mathcal{Z}_2$. To circumvent this requirement, we observe that, compared to $\mathcal{Z}_2$, a model relies on $\mathcal{Z}_1$ should have similar gradients across different domains, since $\mathcal{Z}_1$ is domain-agnostic while $\mathcal{Z}_2$ is domain-specific. Therefore, we propose a simple gradient similarity measure to evaluate how much a model relies on domain-agnostic features. Specifically, given $k$ different domains, $D = \{D_i\}_{i=1}^k$, we denote the mean absolute gradient of $L$ with respect to $\mathbf{z}$ for each domain as

$$\eta_i = \frac{1}{|D_i|} \sum_{(x.y) \in D_i} \mathrm{Abs} \left( \frac{\partial L\left(f_\theta(x), y\right)}{\partial \mathbf{z}} \right),$$ (8)

then the gradient similarity on $D$ is defined as

$$\mathcal{S}(D) = \frac{2 \sum_{1 \le i < j \le k} \mathrm{CosSim}\left(\eta_i, \eta_j\right)}{k(k-1)}, \text{ where } \mathrm{CosSim}\left(\mu, \nu\right) = \frac{\mu \cdot \nu}{\|\mu\|\|\nu\|}.$$ (9)

Intuitively, Equation 9 calculates the cosine similarity between every possible pair of $\eta_i$ and $\eta_j$ that are derived from two different domains, and averages the similarity values together.

## 3 EXPERIMENTS

### 3.1 EXPERIMENTAL SETUP

We evaluate our proposed methods on three domain generalization datasets: PACS (Li et al., 2017) with 4 artistic styles and 7 categories, OfficeHome (Venkateswara et al., 2017) with 4 artistic styles and 65 categories, and Terra Incognita (Beery et al., 2018) with 4 camera locations and 10 categories. As noted by Ye et al. (2022), these datasets exhibit significant diversity shifts and thus are suitable for our evaluation. Visual samples from the PACS dataset are provided in Appendix A.

For simplicity and fair comparison with existing work (Ye et al., 2022; Gulrajani & Lopez-Paz, 2020), we employ ResNet-18 (He et al., 2016) as the backbone architecture for all three datasets. In addition, to further substantiate the effectiveness of our method, we also conduct experiments on the PACS dataset using ResNet-50 as the backbone. There are three model selection strategies commonly used for domain generalization tasks: training-domain validation, test-domain validation, and leave-one-domain-out validation. In our experiments, we adopt training-domain validation following the evaluation protocol used by Ye et al. (2022), Gulrajani & Lopez-Paz (2020) and Wang et al. (2022). All experiments are conducted using Pytorch on Tesla V100 GPUs.

### 3.2 DOMAIN GENERALIZATION

In this section, we present the test accuracy of various methods on the unseen test domains to evaluate their domain generalization performance. For the PACS dataset, we train the model for 7000 steps and perform eight rounds of random searching procedures for weight initialization, dataset division, and hyperparameter combinations. For the OfficeHome and Terra Incognita datasets, we train the model with 5100 steps and perform three rounds of random searching of the basic hyperparameters. For all datasets, we conduct three independent training runs to obtain the average results. Appendix B provides more details about the search space. We also evaluate the sensitivity of the proposed method to different loss weights $(\alpha, 1 - \alpha)$ in Appendix D.2, and we observe that a higher weight for the main classifier works better in practice. As such, we primarily use loss weights $(\alpha, 1 - \alpha) \in \{(0.9, 0.1), (0.99, 0.01)\}$ for other experiments.

**PACS** The PACS dataset comprises four training domain combinations $\{(C, P, S), (A, P, S), (A, C, S), (A, C, P)\}$ that correspond to four test domains $\{A, C, P, S\}$. We set $\epsilon \in \{0.001, 0.005, 0.01, 0.05, 0.1, 0.2\}$ for different domain combinations in the PACS dataset. Appendix C.1 contains detailed accuracy results for various combinations of $\epsilon$ and $(\alpha, 1 - \alpha)$. We also vary the initial learning rate for the main and domain classifiers, as shown in Appendix C.2. Table 1 presents the best results on the PACS dataset. Compared to empirical risk minimization

(ERM), DFP improves the accuracy on every test domain, resulting in a $2.9\%$ increase in the average accuracy. It is worth noting that the improvement varies across different domains, and is more significant on difficult domains such as $A$, $C$, and $S$.

Table 1: Test accuracies of ERM and DFP on PACS.

| PACS | A | C | P | S | Avg |
|---|---|---|---|---|---|
| ERM | $77.9 \pm 0.9$ | $72.7 \pm 0.3$ | $95.8 \pm 0.2$ | $74.1 \pm 0.4$ | 80.1 |
| DFP (Ours) | $\mathbf{80.6 \pm 0.9}$ | $\mathbf{75.8 \pm 1.4}$ | $\mathbf{96.2 \pm 0.3}$ | $\mathbf{79.3 \pm 1.0}$ | $\mathbf{83.0 \uparrow 2.9}$ |

To compare the performance of different model sizes, we also repeat the experiments on the PACS dataset using Resnet-50 as the backbone. Note that the model is only trained for 2100 steps, and the hyperparameters are not extensively tuned for this case. Detailed results are available in Appendix C.4, with the best results are highlighted in Table 2. Compared to ERM using the same backbone, DFP improves the average test accuracy by $1.8\%$.

Table 2: Test accuracies of ERM and DFP with Resnet-50.

| Method | A | C | P | S | Avg |
|---|---|---|---|---|---|
| ERM | $82.6 \pm 1.1$ | $79.7 \pm 0.4$ | $97.3 \pm 0.2$ | $74.7 \pm 1.3$ | 83.6 |
| DFP (Ours) | $\mathbf{84.7 \pm 1.6}$ | $\mathbf{79.7 \pm 1.2}$ | $97.0 \pm 0.1$ | $\mathbf{80.3 \pm 0.8}$ | $\mathbf{85.4 \uparrow 1.8}$ |

**OfficeHome** In line with PACS, the OfficeHome dataset also has four training domain combinations, and four test domains $\{A, C, P, R\}$. We set the $\epsilon \in \{0.1, 0.01, 0.001\}$ and the loss weights $(\alpha, 1 - \alpha) = (0.9, 0.1)$ for the OfficeHome dataset. The complete results are presented in Appendix C.3. As shown in Table 3, our method can enhance the test accuracy of the OfficeHome dataset by $1.4\%$ when compared to the ERM. The basic test accuracies are not particularly high for all test domains, and the increasing effect of DFP leads in less difference between test domains when compared to the results of PACS.

Table 3: Test accuracies of ERM and DFP on OfficeHome.

| OfficeHome | A | C | P | R | Avg |
|---|---|---|---|---|---|
| ERM | $54.8 \pm 0.2$ | $49.8 \pm 0.4$ | $72.3 \pm 0.4$ | $73.4 \pm 0.1$ | 62.6 |
| DFP (Ours) | $\mathbf{57.4 \pm 0.9}$ | $\mathbf{51.0 \pm 0.4}$ | $\mathbf{73.4 \pm 0.1}$ | $\mathbf{74.3 \pm 0.4}$ | $\mathbf{64.0 \uparrow +1.4}$ |

**Terra Incognita** The Terra Incognita dataset also includes four training domain combinations, as well as four test domain types $\{L100, L38, L43, L46\}$. And we set the combination of $\epsilon$ and the loss weights $(\epsilon, \alpha, 1 - \alpha) \in \{(0.1, 0.9, 0.1), (0.01, 0.9, 0.1), (0.01, 0.99, 0.01)\}$ for the Terra Incognita dataset. The complete results are presented in Appendix C.3. As shown in Table 4, our method can enhance the test accuracy of the Terra Incognita dataset by $3.5\%$ when compared to the ERM. The Terra Incognita dataset contains images of wild animals captured by camera traps in a variety of natural environments, simulating a real-world scenario. Despite the fact that DFP has improved test accuracies across domains, the baseline and DFP results are not particularly impressive. These results show that OOD generalization is more difficult in photos with more intricate and realistic backgrounds, such as The Terra Incognita dataset.

Table 4: Test accuracies of ERM and DFP on Terra Incognita.

| TerraIncognita | L100 | L38 | L43 | L46 | Avg |
|---|---|---|---|---|---|
| ERM | $44.4 \pm 4.2$ | $38.0 \pm 2.8$ | $50.5 \pm 1.1$ | $36.3 \pm 0.1$ | 42.3 |
| DFP (Ours) | $\mathbf{47.4 \pm 1.9}$ | $\mathbf{39.7 \pm 2.7}$ | $\mathbf{52.6 \pm 0.0}$ | $\mathbf{37.2 \pm 0.9}$ | $\mathbf{44.2 \uparrow +1.9}$ |

**Other comparisons** Drawing on the findings from Ye et al. (2022), we conduct a comparative analysis between our method and some state-of-the-art techniques. Table 5 presents these results, indicating that our approach consistently surpasses several established methods. Additionally, Table 5 includes our own results for ERM Vapnik (1999) and RSC Huang et al. (2020). ERM is a standard baseline for comparisons in domain generalization (DG) challenges. Following benchmark

Table 5: Test accuracy comparison with other methods. Results of the first block are taken from Ye et al. (2022).

| dataset / method | PACS | OfficeHome | TerraIncognita | Avg |
|---|---|---|---|---|
| CORALSun & Saenko (2016) | 81.6 ± 0.6 | 63.8 ± 0.3 | 38.3 ± 0.7 | 61.23 |
| MMDLi et al. (2018b) | 81.7 ± 0.2 | 63.8 ± 0.1 | 38.3 ± 0.4 | 61.27 |
| ERDGZhao et al. (2020) | 80.5 ± 0.5 | 63 ± 0.4 | 41.3 ± 1.2 | 61.60 |
| IGAKoyama & Yamaguchi (2020) | 80.9 ± 0.4 | 63.6 ± 0.2 | 41.3 ±0.8 | 61.93 |
| VRExKrueger et al. (2021) | 81.8 ± 0.1 | 63.5 ± 0.1 | 40.7 ± 0.7 | 62.00 |
| IRMArjovsky et al. (2019) | 81.1 ± 0.3 | 63 ± 0.2 | 42 ± 1.8 | 62.03 |
| SagNetNam et al. (2019) | 81.6 ± 0.4 | 62.7 ± 0.4 | 42.3 ± 0.7 | 62.20 |
| ERMVapnik (1999) | 81.5 ± 0.0 | 63.3 ± 0.2 | 42.6 ± 0.9 | 62.47 |
| RSCHuang et al. (2020) | 82.8 ± 0.4 | 62.9 ± 0.4 | 43.6 ± 0.5 | 63.10 |
| ERM (Our runs) | 80.1 ± 0.2 | 62.6 ± 0.1 | 42.3 ± 1.1 | 62.03 |
| RSC (Our runs) | 82.0 ± 0.5 | 62.8 ± 0.1 | 44.4 ± 0.0 | 63.27 |
| DFP (Ours) | **83.0 ± 0.7** | **64.0 ± 0.3** | **44.2 ± 0.6** | **63.73** |

results, we also evaluate RSC, which outperforms many other techniques. RSC employs gradient characteristics to mute feature representations with the highest gradient, compelling the model to rely on other features for predictions. In contrast, our method aims to isolate domain-related features and introduce perturbations to them.

## 3.3 GRADIENT SIMILARITY

To illustrate the effect of domain feature perturbation, we examine the similarity computation on the PACS dataset. We define four training domain combinations $\{(C, P, S), (A, P, S), (A, C, S), (A, C, P)\}$ and their respective test domains $\{A, C, P, S\}$. We employ the gradients' absolute values to compute the cosine similarity across different dimensions of hidden representations. Gradient similarities across domains may reflect feature similarities they emphasize. For each training domain set, we independently assess the similarity. For the three domains in each set, we determine the average gradient of hidden representations as described in Section 2.3.

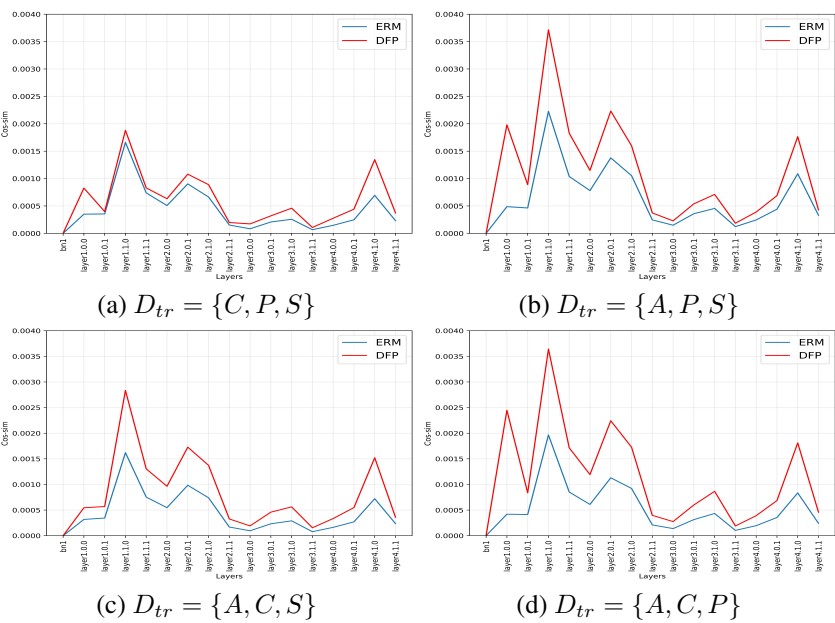

(a) $D_{tr} = \{C, P, S\}$

(b) $D_{tr} = \{A, P, S\}$

(c) $D_{tr} = \{A, C, S\}$

(d) $D_{tr} = \{A, C, P\}$

Figure 2: Gradient similarities at different layers of ERM and DFP.

As illustrated in Figure 2, we calculate the average gradient similarity for each layer subjected to perturbation. To contrast the gradient similarities between DFP and ERM, we select the optimal pre-trained model from each method and perform an additional training step. Evaluating each approach

50 times, we determine the mean similarity value. The findings suggest that DFP can induce a similar gradient change between different source domains.

## 3.4 ABLATION STUDIES

In this section, we present two ablation studies focusing on: (1) a comparison with random perturbations and (2) the optimal position for noise injection. All experiments utilize the ResNet-18 architecture. For both studies, trials are conducted on the PACS dataset with a fixed initial learning rate of $lr=5e$-5 and a training duration of 7000 steps. In each of the three independent training iterations, we perform eight rounds of random hyperparameter search.

**Random perturbation** To evaluate the effectiveness of noise modulation, we train a baseline model with random noise. Experiments using random perturbations are conducted at the same insertion position as DFP. We set the random noise $n \sim \mathcal{N}(0, \sigma^2)$ with $\sigma \in \{0.001, 0.005, 0.01, 0.05, 0.1, 0.2\}$. Table 6 displays the best results of random perturbations. Appendix D.3 shows the detailed results. The average accuracy of random perturbations is $81.3\%$, and the results suggest that our proposed method outperforms random perturbations by $1.7\%$, which is lower than DFP.

Table 6: Test accuracies of ERM with random perturbations.

|  | A | C | P | S | Avg |
|---|---|---|---|---|---|
| best | $79.6 \pm 0.5$ | $75.3 \pm 0.9$ | $95.4 \pm 0.2$ | $74.8 \pm 0.7$ | 81.3 |

**Noise injection point** We compare the effect of two different noise injection points, pre-activation and activation. We set the $\epsilon \in \{0.001, 0.005, 0.01, 0.05, 0.1, 0.2\}$ and the loss weights $(\alpha, 1 - \alpha) \in \{(0.9, 0.1), (0.99, 0.01)\}$. The Appendix D.1 presents more detailed results for various combinations of $\epsilon$ and $(\alpha, 1 - \alpha)$. Table 7 shows the best results, and there is no notable difference between the two cases.

Table 7: Test accuracies of DFP with different injection points.

|  | A | C | P | S | Avg |
|---|---|---|---|---|---|
| pre-act | $80.6 \pm 0.9$ | $74.7 \pm 0.6$ | $95.9 \pm 0.4$ | $77.7 \pm 0.8$ | 82.2 |
| after-act | $79.9 \pm 0.4$ | $76.0 \pm 1.0$ | $96.0 \pm 0.3$ | $77.2 \pm 0.9$ | 82.2 |

## 4 RELATED WORK

In this section, we review existing approaches to domain generalization, and discuss their connections to our work.

**Data augmentation** Data augmentation is a prevalent technique to enhance data diversity in various machine learning tasks (Shorten & Khoshgoftaar, 2019; Yan et al., 2020). In the context of domain generalization, existing methods often emphasize feature-level augmentation (Li et al., 2021). Mixstyle (Zhou et al., 2021) generates new styles by blending feature statistics from two instances with random convex weights. Xu et al. (2021) employs Fourier-based data augmentation, consolidating information from multiple source domains. Other approaches, including domain randomization (Tobin et al., 2017; Huang et al., 2021; Fan et al., 2022), introduce perturbations to data statistics to simulate diverse visual styles. We note that DFP is loosely related to domain randomization in that it also aims to randomize domain-specific features. Nevertheless, the perturbations introduced by DFP are applied to intermediate representations, and are automatically generated with a domain classifier.

**Invariant learning** Another potential challenge for OOD generalization is the tendency of DNNs to overly specialize in extracting features that are useful for the training data. As such, several strategies have been developed to cultivate robust representations. These include invariant representation (Parascandolo et al., 2020), meta-learning (Zhang et al., 2020; Li et al., 2018a; Bui

et al., 2021), transfer learning (Blanchard et al., 2021), representation disentanglement (Zhang et al., 2022), model calibration (Wald et al., 2021), and causal learning (Mahajan et al., 2021; Lv et al., 2022). Contrary to learning invariant representation, Shi et al. (2021) and Rame et al. (2022) focus on gradient invariance during training. While DFP does not explicitly learn invariant representation, its carefully crafted perturbation encourages more dependence on invariant representation as discussed in Section 2.2. This distinction makes DFP potentially better suited for multitask learning, which requires the extraction of diverse features.

In addition to data augmentation and invariant learning, more general regularization techniques have been explored to tackle the OOD challenge, including ensemble learning (Arpit et al., 2022; Segu et al., 2023; Li et al., 2022), sharpness-aware optimization (Cha et al., 2021), adversarial training (Ganin et al., 2016; Lee et al., 2019; Yi et al., 2021; Wang et al., 2022), and several others (Sagawa et al., 2019; Kim et al., 2021; Chen et al., 2022). Notably, among the adversarial training-based approaches, some also apply perturbations to intermediate representations (Sankaranarayanan et al., 2018; You et al., 2019). Compared to these approaches, DFP generate the perturbations without adversarial training, resulting in a simpler and more stable training procedure.

## 5 CONCLUSION

In this work, we proposed a novel approach to domain generalization, named domain feature perturbation (DFP). DFP incorporates a domain classifier to produce perturbations for domain-specific features, aiming to reduce the dependence of the model on such features. Furthermore, we conducted extensive experiments on multiple domain generalization datasets, demonstrating the effectiveness of DFP, as well as competitive or better OOD performance than state-of-the-art methods.

Our method has several limitations. First, DFP relies on two forward and backward passes in each training iteration, rendering it slower at training time than simple approaches such as empirical risk minimization. Second, using the gradient of the domain classifier, we can only approximately identify domain-specific features, making the regularization effect of DFP less precise. Therefore, future research might present methods for the more efficient and precise extraction of these features.

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

## A    ILLUSTRATION OF DATASETS

**Datasets**    We evaluate the efficacy of our approach across three representative datasets for tasks involving OOD generalization: PACS (Li et al., 2017) with 4 domains, $\{photos, art, cartoons, sketches\}$ and 7 classes; OfficeHome (Venkateswara et al., 2017) with 4 domains $\{art, clipart, product, real\}$ and 65 classes; Terra Incognita (Beery et al., 2018) with four of the camera locations $\{L100, L38, L43, L46\}$ and 10 classes. According to the Ye et al. (2022), these datasets are characterized by substantial diversity shifts, aligning with our objective of leveraging domain-specific information. Figure 3 shows the illustration of images in the PACS dataset, with 4 domains and 7 classes. We also evaluate the cosine-similarity of images between domains in the PACS dataset, and Figure 4 shows the results.

## B    HYPER-PARAMETER SEARCH SPACE

**General hyperparameters**    We set the initial learning rate to $lr=5e\text{-}5$ for all datasets. We also set the dropout rate for the main object classifier to zero. Table 8 shows other basic hyperparameter values for ERM and RSC.

**DFP hyperparameters**    The major noise-related hyperparameter for our proposed Domain Feature Perturbation (DFP) approach for domain generalization is $\epsilon$, which regulates the standard deviation $\sigma = [|\nabla_z l_d|/\|\nabla_z l_d\|_p] \cdot \epsilon$ with $p=\infty$. Furthermore, the loss weights $\alpha$ and $1-\alpha$ are flexible in order to manage the performance of the two classifiers. We experiment with numerous $\epsilon$ and $(\alpha, 1-\alpha)$ combinations for different datasets. Table 9 depicts the grid search space of these hyper-parameters.

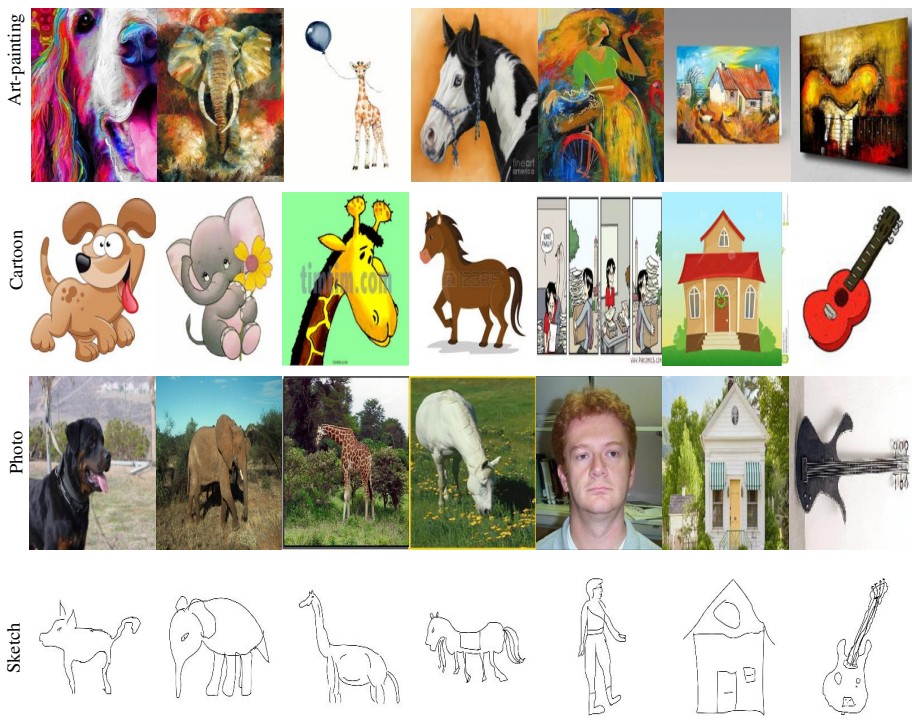

(a) Dog  (b) Elephant (c) Giraffe  (d) Horse  (e) Person  (f) House  (g) Guitar

Figure 3: Illustration of images in the PACS dataset.

Table 8: Basic hyper-parameters of experiments.

| Parameter | Value |
|---|---|
| learning rate | $5e\text{-}5$ |
| weight_decay | $10^{uniform(-6,-2)}$ |
| resnet_dropout | 0 |
| data_augmentation | true |
| batch_size | default $= 32$, $2^{uniform(3,5.5)}$ |
| rsc_f_drop_factor | default $= 1/3$, $1^{uniform(0,0.5)}$ |
| rsc_b_drop_factor | default $= 1/3$, $1^{uniform(0,0.5)}$ |

Table 9: DFP hyper-parameters grid search space.

| Parameter | Value |
|---|---|
| $\epsilon$ | [0.2,0.1,0.05,0.01,0.005,0.001] |
| $(\alpha, 1-\alpha)$ | [(0.99,0.01), (0.9,0.1)] |
| p | [$\infty$,1,2] |

**Model selection method**   We use the training-domain validation method to select the model. We divide each training domain into training and validation subsets for the training-domain validation technique. We train models with the training subsets and select the model with the highest accuracy based on the union of the validation subsets. This technique implies that the distributions of the training and test cases are similar.

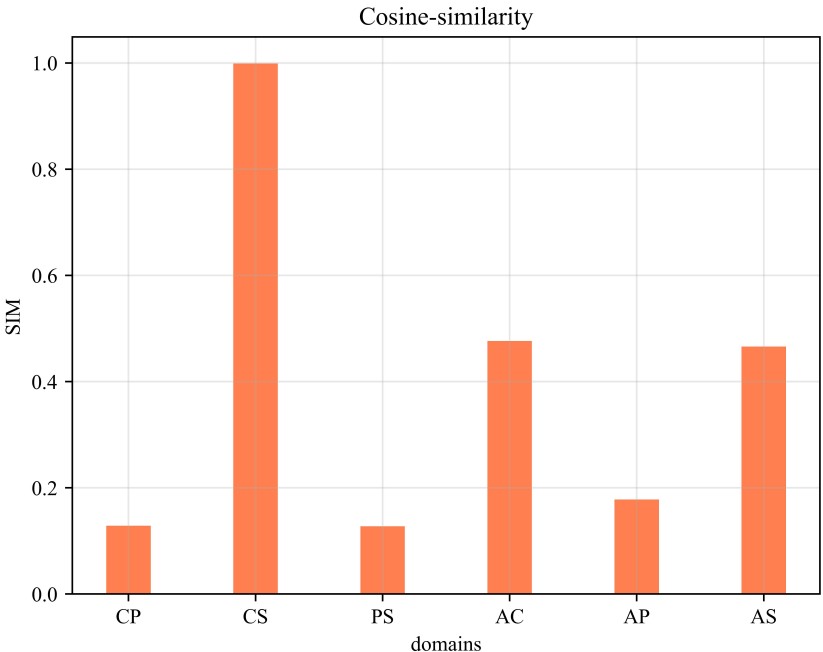

Figure 4: Cosine similarity of images between different domains in the PACS dataset.

## C MORE RESULTS OF DFP

### C.1 DIFFERENT PERTURBATION LEVELS

All of the results in this section are based on the Resnet-18 structure and are tested on the PACS dataset. Table 10 displays the complete results of several hyperparameter combinations in DFP.

Table 10: Test accuracies of DFP with resnet-18.

| $(\alpha, 1-\alpha)$ | $\epsilon$ | A | C | P | S | Avg |
|---|---|---|---|---|---|---|
| (0.9,0.1) | 0.2 | $78.3 \pm 0.6$ | $72.8 \pm 0.6$ | $95.5 \pm 0.4$ | $76.5 \pm 1.0$ | |
| (0.99,0.01) | 0.2 | $77.6 \pm 1.1$ | $74.4 \pm 0.5$ | $95.5 \pm 0.1$ | $72.8 \pm 1.2$ | |
| (0.9,0.1) | 0.1 | $77.4 \pm 0.1$ | $73.6 \pm 0.2$ | $95.6 \pm 0.1$ | $76.4 \pm 0.7$ | |
| (0.99,0.01) | 0.1 | $78.3 \pm 0.6$ | $74.2 \pm 2.3$ | $\mathbf{95.9 \pm 0.4}$ | $76.7 \pm 1.2$ | |
| (0.9,0.1) | 0.05 | $\mathbf{80.6 \pm 0.9}$ | $72.2 \pm 2.2$ | $95.0 \pm 0.3$ | $74.6 \pm 0.8$ | |
| (0.99,0.01) | 0.05 | $75.9 \pm 1.0$ | $\mathbf{74.7 \pm 0.6}$ | $94.9 \pm 0.2$ | $\mathbf{77.7 \pm 0.8}$ | |
| (0.9,0.1) | 0.01 | $79.3 \pm 1.5$ | $72.4 \pm 1.0$ | $95.2 \pm 0.6$ | $72.4 \pm 1.3$ | |
| (0.99,0.01) | 0.01 | $76.6 \pm 1.8$ | $72.9 \pm 0.1$ | $95.7 \pm 0.5$ | $70.9 \pm 1.5$ | |
| (0.9,0.1) | 0.005 | $77.0 \pm 0.5$ | $72.5 \pm 0.2$ | $95.6 \pm 0.4$ | $75.6 \pm 1.2$ | |
| (0.99,0.01) | 0.005 | $78.7 \pm 1.8$ | $74.2 \pm 0.8$ | $95.7 \pm 0.2$ | $76.2 \pm 1.5$ | |
| (0.9,0.1) | 0.001 | $79.0 \pm 1.4$ | $73.7 \pm 0.2$ | $95.7 \pm 0.4$ | $77.3 \pm 1.4$ | |
| (0.99,0.01) | 0.001 | $75.4 \pm 0.9$ | $73.3 \pm 0.6$ | $95.6 \pm 0.3$ | $77.0 \pm 1.0$ | |
| (0.9,0.1) | 0.0005 | $78.0 \pm 0.8$ | $70.8 \pm 1.0$ | $95.4 \pm 0.2$ | $73.6 \pm 0.9$ | |
| (0.99,0.01) | 0.0005 | $77.4 \pm 0.6$ | $73.7 \pm 0.8$ | $95.7 \pm 0.3$ | $77.7 \pm 1.1$ | |
| | best | $\mathbf{80.6 \pm 0.9}$ | $\mathbf{74.7 \pm 0.6}$ | $\mathbf{95.9 \pm 0.4}$ | $\mathbf{77.7 \pm 0.8}$ | 82.2 |

### C.2 DIFFERENT LEARNING RATES

We train the model with the Resnet-18 structure and test it on the PACS dataset. We keep the main classifier's initial learning rate at 5e-5 and experiment with alternative learning rate settings for the domain classifier. The entire results of different learning rates of the domain classifier in DFP are shown in Table 11 and Table 12.

Table 11: Test accuracies of DFP with different learning rates (1e−4).

| $(\alpha, 1-\alpha)$ | $\epsilon$ | A | C | P | S | Avg |
|---|---|---|---|---|---|---|
| (0.9,0.1) | 0.1 | $78.1 \pm 1.4$ | $73.0 \pm 2.3$ | $95.4 \pm 0.2$ | $75.2 \pm 0.9$ | |
| (0.99,0.01) | 0.1 | $76.3 \pm 1.1$ | $73.4 \pm 1.9$ | $95.7 \pm 0.2$ | $75.3 \pm 1.7$ | |
| (0.9,0.1) | 0.05 | $\mathbf{80.1 \pm 0.5}$ | $74.1 \pm 0.7$ | $95.3 \pm 0.1$ | $76.5 \pm 0.1$ | |
| (0.99,0.01) | 0.05 | $76.7 \pm 1.0$ | $72.1 \pm 1.5$ | $95.3 \pm 0.3$ | $75.4 \pm 0.9$ | |
| (0.9,0.1) | 0.01 | $80.0 \pm 0.3$ | $73.7 \pm 0.2$ | $95.5 \pm 0.2$ | $76.6 \pm 0.8$ | |
| (0.99,0.01) | 0.01 | $78.2 \pm 0.6$ | $73.2 \pm 0.5$ | $95.6 \pm 0.5$ | $75.6 \pm 0.4$ | |
| (0.9,0.1) | 0.005 | $76.4 \pm 1.4$ | $73.4 \pm 0.5$ | $95.3 \pm 0.4$ | $76.9 \pm 1.3$ | |
| (0.99,0.01) | 0.005 | $78.2 \pm 0.7$ | $74.3 \pm 0.7$ | $95.5 \pm 0.1$ | $\mathbf{77.2 \pm 0.5}$ | |
| (0.9,0.1) | 0.001 | $78.8 \pm 1.3$ | $\mathbf{74.5 \pm 0.6}$ | $95.6 \pm 0.4$ | $75.8 \pm 0.4$ | |
| (0.99,0.01) | 0.001 | $78.8 \pm 0.8$ | $74.0 \pm 0.7$ | $\mathbf{96.2 \pm 0.3}$ | $74.8 \pm 2.5$ | |
| | best | $\mathbf{80.1 \pm 0.1}$ | $\mathbf{74.5 \pm 0.6}$ | $\mathbf{96.2 \pm 0.3}$ | $\mathbf{77.2 \pm 0.5}$ | 82 |

Table 12: Test accuracies of DFP with different learning rates (1e−5).

| $(\alpha, 1-\alpha)$ | $\epsilon$ | A | C | P | S | Avg |
|---|---|---|---|---|---|---|
| (0.9,0.1) | 0.1 | $76.4 \pm 0.4$ | $74.4 \pm 0.6$ | $95.5 \pm 0.4$ | $75.7 \pm 1.2$ | |
| (0.99,0.01) | 0.1 | $77.8 \pm 1.2$ | $75.5 \pm 0.2$ | $95.3 \pm 0.3$ | $77.8 \pm 0.9$ | |
| (0.9,0.1) | 0.05 | $77.5 \pm 1.3$ | $73.5 \pm 1.1$ | $95.1 \pm 0.3$ | $77.2 \pm 0.3$ | |
| (0.99,0.01) | 0.05 | $77.2 \pm 0.4$ | $74.7 \pm 1.0$ | $95.2 \pm 0.1$ | $75.3 \pm 0.7$ | |
| (0.9,0.1) | 0.01 | $77.5 \pm 1.4$ | $74.1 \pm 0.8$ | $94.2 \pm 0.6$ | $74.1 \pm 1.2$ | |
| (0.99,0.01) | 0.01 | $76.9 \pm 1.3$ | $\mathbf{75.8 \pm 1.4}$ | $95.0 \pm 0.4$ | $76.4 \pm 1.5$ | |
| (0.9,0.1) | 0.005 | $\mathbf{78.9 \pm 1.3}$ | $73.5 \pm 0.9$ | $95.2 \pm 0.1$ | $76.2 \pm 1.6$ | |
| (0.99,0.01) | 0.005 | $78.5 \pm 0.9$ | $74.5 \pm 1.1$ | $95.7 \pm 0.3$ | $\mathbf{78.3 \pm 1.2}$ | |
| (0.9,0.1) | 0.001 | $76.3 \pm 0.9$ | $72.1 \pm 0.9$ | $\mathbf{96.1 \pm 0.3}$ | $75.7 \pm 0.3$ | |
| (0.99,0.01) | 0.001 | $75.7 \pm 1.9$ | $74.9 \pm 0.5$ | $95.6 \pm 0.5$ | $76.8 \pm 0.2$ | |
| | best | $\mathbf{78.9 \pm 1.3}$ | $\mathbf{75.8 \pm 1.4}$ | $\mathbf{96.1 \pm 0.3}$ | $\mathbf{78.3 \pm 1.2}$ | 82.275 |

## C.3 DIFFERENT DATASETS

The OfficeHome dataset has four training domain combinations $\{(C,P,R),(A,P,R),(A,C,R),(A,C,P)\}$, and four test domain $\{A,C,P,R\}$. The Terra Incognita dataset also includes four training domain combinations $\{(L38,L43,L46),(L100,L43,L46),(L100,L38,L46),(L100,L38,L43)\}$, as well as four test domain types $\{(L38,L43,L46),(L100,L38,L43)\}$. Table 13 displays the outcomes of the OfficeHome dataset with various hyperparameter combinations. Table 14 illustrates the Terra Incognita dataset results with various hyperparameter combinations.

Table 13: Test accuracies of DFP on OfficeHome.

| $(\alpha, 1-\alpha)$ | $\epsilon$ | A | C | P | S | Avg |
|---|---|---|---|---|---|---|
| (0.9,0.1) | 0.1 | $57.4 \pm 0.9$ | $50.1 \pm 0.4$ | $73.4 \pm 0.1$ | $74.3 \pm 0.4$ | |
| (0.9,0.1) | 0.01 | $56.3 \pm 0.2$ | $50.2 \pm 0.4$ | $73.0 \pm 0.3$ | $74.3 \pm 0.4$ | |
| (0.9,0.1) | 0.001 | $56.0 \pm 0.1$ | $51.0 \pm 0.4$ | $72.9 \pm 0.3$ | $74.1 \pm 0.2$ | |
| | best | $\mathbf{57.4 \pm 0.9}$ | $\mathbf{51.0 \pm 0.4}$ | $\mathbf{73.4 \pm 0.1}$ | $\mathbf{74.3 \pm 0.4}$ | 64.0 |

Table 14: Test accuracies of DFP on Terra Incognita.

| $(\alpha, 1-\alpha)$ | $\epsilon$ | A | C | P | S | Avg |
|---|---|---|---|---|---|---|
| (0.9,0.1) | 0.1 | $45.3 \pm 4.1$ | $36.4 \pm 1.5$ | $52.6 \pm 0.0$ | $33.6 \pm 1.1$ | |
| (0.9,0.1) | 0.01 | $44.4 \pm 4.2$ | $36.7 \pm 2.6$ | $50.5 \pm 1.1$ | $36.8 \pm 0.1$ | |
| (0.99,0.01) | 0.01 | $47.4 \pm 1.9$ | $39.7 \pm 2.7$ | $51.2 \pm 0.4$ | $37.2 \pm 0.9$ | |
| | best | $\mathbf{47.4 \pm 1.9}$ | $\mathbf{39.7 \pm 2.7}$ | $\mathbf{52.6 \pm 0.0}$ | $\mathbf{37.2 \pm 0.9}$ | 44.2 |

## C.4 DIFFERENT MODEL ARCHITECTURES

Table 15 displays results based on the Resnet-50 structure with various hyperparameter combinations. We test the model on the PACS dataset.

Table 15: Test accuracies of ERM and DFP with Resnet-50.

| $(\alpha, 1-\alpha)$ | $\epsilon$ | A | C | P | S | Avg |
|---|---|---|---|---|---|---|
| | ERM | $82.6 \pm 1.1$ | $79.7 \pm 0.4$ | $97.3 \pm 0.2$ | $74.7 \pm 1.3$ | 83.575 |
| $(0.9, 0.1)$ | 0.001 | $84.3 \pm 0.9$ | $76.9 \pm 0.3$ | $97.0 \pm 0.3$ | $75.5 \pm 1.0$ | |
| | 0.005 | $82.0 \pm 1.3$ | $78.2 \pm 0.8$ | $96.8 \pm 0.2$ | $74.1 \pm 2.4$ | |
| | 0.01 | $81.5 \pm 0.8$ | $77.9 \pm 0.9$ | $96.5 \pm 0.1$ | $\mathbf{80.3 \pm 0.8}$ | |
| | 0.05 | $84.1 \pm 1.2$ | $79.6 \pm 0.4$ | $96.6 \pm 0.4$ | $77.8 \pm 0.3$ | |
| | 0.1 | $\mathbf{84.7 \pm 1.6}$ | $79.6 \pm 0.3$ | $96.7 \pm 0.2$ | $74.1 \pm 3.1$ | |
| $(0.99, 0.01)$ | 0.001 | $82.8 \pm 0.8$ | $79.5 \pm 0.4$ | $\mathbf{97.0 \pm 0.1}$ | $79.9 \pm 1.1$ | |
| | 0.005 | $80.2 \pm 1.2$ | $75.5 \pm 1.0$ | $96.7 \pm 0.2$ | $77.6 \pm 0.5$ | |
| | 0.01 | $82.7 \pm 0.5$ | $77.4 \pm 1.2$ | $96.6 \pm 0.2$ | $78.0 \pm 0.3$ | |
| | 0.05 | $83.4 \pm 0.8$ | $\mathbf{79.7 \pm 1.2}$ | $96.8 \pm 0.2$ | $76.8 \pm 1.7$ | |
| | 0.1 | $84.0 \pm 0.5$ | $79.1 \pm 0.4$ | $96.8 \pm 0.2$ | $74.8 \pm 2.3$ | |
| | best | $\mathbf{84.7 \pm 1.6}$ | $\mathbf{79.7 \pm 1.2}$ | $\mathbf{97.0 \pm 0.1}$ | $\mathbf{80.3 \pm 0.8}$ | 85.425 |

# D MORE RESULTS OF ABLATION STUDIES

## D.1 NOISE INJECTION POINT

Table 16 shows the outcomes with different positions for adding the perturbations. The results are based on ResNet-18 model and the PACS dataset.

Table 16: Test accuracies of ERM with random perturbation.

| $(\alpha, 1-\alpha)$ | $\epsilon$ | A | C | P | S | Avg |
|---|---|---|---|---|---|---|
| $(0.9, 0.1)$ | 0.2 | $78.5 \pm 0.7$ | $72.6 \pm 0.9$ | $95.6 \pm 0.0$ | $72.6 \pm 1.8$ | |
| | 0.1 | $76.5 \pm 1.0$ | $70.9 \pm 1.3$ | $94.9 \pm 0.4$ | $75.4 \pm 0.5$ | |
| | 0.05 | $78.1 \pm 0.2$ | $73.5 \pm 0.9$ | $95.2 \pm 0.7$ | $77.2 \pm 0.9$ | |
| | 0.01 | $78.5 \pm 1.1$ | $73.3 \pm 1.4$ | $95.3 \pm 0.2$ | $76.8 \pm 1.4$ | |
| | 0.005 | $77.4 \pm 0.9$ | $73.5 \pm 0.8$ | $95.1 \pm 0.4$ | $75.4 \pm 0.7$ | |
| | 0.001 | $76.5 \pm 0.1$ | $71.4 \pm 1.2$ | $94.9 \pm 0.5$ | $75.5 \pm 0.6$ | |
| $(0.99, 0.01)$ | 0.2 | $74.5 \pm 0.9$ | $72.6 \pm 0.3$ | $95.8 \pm 0.3$ | $74.8 \pm 0.6$ | |
| | 0.1 | $78.5 \pm 0.9$ | $76.0 \pm 1.0$ | $95.3 \pm 0.2$ | $73.0 \pm 3.3$ | |
| | 0.05 | $78.1 \pm 1.2$ | $73.7 \pm 1.1$ | $95.3 \pm 0.5$ | $75.4 \pm 0.2$ | |
| | 0.01 | $76.0 \pm 0.5$ | $75.6 \pm 1.0$ | $95.8 \pm 0.2$ | $76.3 \pm 1.0$ | |
| | 0.005 | $79.8 \pm 1.0$ | $72.8 \pm 0.7$ | $95.4 \pm 0.1$ | $73.0 \pm 2.7$ | |
| | 0.001 | $79.9 \pm 0.4$ | $73.4 \pm 2.1$ | $96.0 \pm 0.3$ | $74.1 \pm 2.2$ | |
| | best | $\mathbf{79.9 \pm 0.4}$ | $\mathbf{76.0 \pm 1.0}$ | $\mathbf{96.0 \pm 0.3}$ | $\mathbf{77.2 \pm 0.9}$ | 82.3 |

## D.2 SENSITIVITY ANALYSIS

To scrutinize the sensitivity of our proposed approach to different loss weights, we conduct experiments on the PACS dataset, maintaining consistent parameters such as the initial learning rate $lr$=5$e$-5 and a training duration of 7000 steps. In addition, we run one random search of basic hyperparameters for each of the five independent training series. We set the random noise $n \sim \mathcal{N}(0, \sigma^2)$ with $\sigma = 0.05$. And the loss weights $(\alpha, 1 - \alpha) \in \{(0.5, 0.5), (0.6, 0.4), (0.7, 0.3), (0.8, 0.2), (0.9, 0.1), (0.99, 0.01)\}$. The results are shown in Table 17 and Figure 5.According to the results, we primarily utilize loss weights $(\alpha, 1 - \alpha) \in \{(0.9, 0.1), (0.99, 0.01)\}$ for other investigations.

Table 17: Test accuracies of DFP with different loss weights.

| $\epsilon$ | $(\alpha, 1-\alpha)$ | A | C | P | S | Avg |
|---|---|---|---|---|---|---|
| 0.05 | $(0.99, 0.01)$ | $79.1 \pm 0.8$ | $73.9 \pm 0.5$ | $95.5 \pm 0.4$ | $75.2 \pm 1.4$ | 80.9 |
| | $(0.9, 0.1)$ | $77.4 \pm 0.6$ | $72.1 \pm 0.8$ | $95.4 \pm 0.1$ | $77.1 \pm 0.9$ | 80.5 |
| | $(0.8, 0.2)$ | $77.7 \pm 0.6$ | $73.4 \pm 0.8$ | $95.5 \pm 0.2$ | $74.1 \pm 0.9$ | 80.2 |
| | $(0.7, 0.3)$ | $77.6 \pm 0.9$ | $71.4 \pm 1.2$ | $95.5 \pm 0.2$ | $73.1 \pm 1.3$ | 79.4 |
| | $(0.6, 0.4)$ | $78.3 \pm 0.7$ | $70.7 \pm 0.6$ | $95.3 \pm 0.3$ | $74.3 \pm 0.9$ | 79.5 |
| | $(0.5, 0.5)$ | $76.6 \pm 0.7$ | $70.0 \pm 1.1$ | $95.9 \pm 0.3$ | $75.3 \pm 1.0$ | 79.7 |

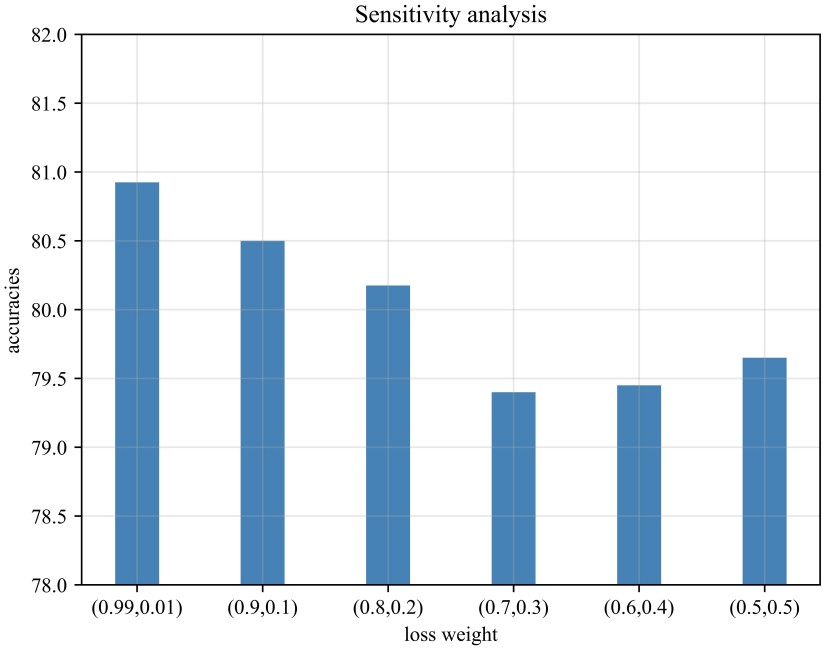

Figure 5: Accuracy results of different loss weights.

## D.3 RANDOM PERTURBATIONS

We set the random noise $n \sim \mathcal{N}(\mu, \sigma^2)$ with $\sigma \in [0.001, 0.005, 0.01, 0.05, 0.1, 0.2]$. And the results are shown in Table 18.

Table 18: Test accuracies of ERM with random perturbations.

| $\epsilon$ | A | C | P | S | Avg |
|---|---|---|---|---|---|
| 0.001 | $74.4 \pm 0.3$ | $74.3 \pm 0.3$ | $95.4 \pm 0.2$ | $74.7 \pm 1.2$ | |
| 0.005 | $75.6 \pm 1.3$ | $\mathbf{75.3 \pm 0.9}$ | $\mathbf{95.4 \pm 0.5}$ | $72.6 \pm 1.5$ | |
| 0.01 | $76.5 \pm 1.3$ | $73.0 \pm 1.2$ | $95.1 \pm 0.1$ | $\mathbf{74.8 \pm 0.7}$ | |
| 0.05 | $77.4 \pm 1.2$ | $72.5 \pm 1.2$ | $95.3 \pm 0.1$ | $74.2 \pm 1.8$ | |
| 0.1 | $\mathbf{79.6 \pm 0.5}$ | $67.8 \pm 0.5$ | $91.6 \pm 0.6$ | $62.7 \pm 2.0$ | |
| 0.2 | $70.6 \pm 0.2$ | $53.8 \pm 0.8$ | $83.6 \pm 0.9$ | $46.4 \pm 5.3$ | |
| best | $\mathbf{79.6 \pm 0.5}$ | $\mathbf{75.3 \pm 0.9}$ | $\mathbf{95.4 \pm 0.2}$ | $\mathbf{74.8 \pm 0.7}$ | $\mathbf{81.3}$ |

