# OpenReview forum: "Domain Feature Perturbation for Domain Generalization"
_ICLR.cc/2024/Conference — ICLR 2024 Conference Withdrawn Submission_

### Official Review · Reviewer_QF8t · 2023-10-25

**Soundness:** 2 fair
**Presentation:** 3 good
**Contribution:** 2 fair
**Rating:** 3
**Confidence:** 5

**Summary:**

The paper aims to reduce the dependence of the model on domain-specific features in DG, and proposes a simple method named domain feature perturbation(DFP). The method only incorporates a domain classifier to produce perturbations for domain-specific features. Finally,
some experiments have been conducted to verify the effectiveness of the proposed method.

**Strengths:**

1.The paper is well written, and the assumptions are well described and the theories are solid.

2.The proposed method is very simple to understand, and the experiment results show the improvement compared to ERM.

**Weaknesses:**

1.Lack of comparison with the latest methods, the latest comparison method in Table 5 is VREx published in 2021, which can incorporate recent DG methods such as Fishr[1] and DCG[2]. Moreover, the effectiveness of the method proposed in the paper is not as good as most of current methods.


2.The ablation experiments are not enough to show the contribution of each component in the method, and the experimental settings and comparison should be more detailed.

3.The test accuracy enhancement on Terra Incognita dataset is inconsistent with the data in table 4.

[1] Alexandre Rame, Corentin Dancette, and Matthieu Cord. Fishr: Invariant gradient variances for out-of-distribution generalization. ICML 2022.

[2] Lv, Fangrui, Jian Liang, Shuang Li, Jinming Zhang, and Di Liu. Improving Generalization with Domain Convex Game. CVPR 2023.

**Questions:**

Please refer to weaknesses.

---

### Official Review · Reviewer_U8c9 · 2023-10-30

**Soundness:** 2 fair
**Presentation:** 2 fair
**Contribution:** 2 fair
**Rating:** 5
**Confidence:** 4

**Summary:**

The paper introduces Domain Feature Perturbation (DFP), a novel technique to enhance the performance of Deep Neural Networks (DNNs) in out-of-distribution (OOD) scenarios. This method trains a domain classifier alongside the main prediction model, using a shared backbone network. The DFP approach involves perturbing the multi-layer representation of the prediction model with noise, modulated by the gradient of the domain classifier. This design intends to decrease the main model's reliance on domain-specific features, shifting focus to more general, domain-agnostic features. The paper highlights its ease of implementation and effectiveness in improving performance across several domain generalization benchmarks.

**Strengths:**

+ The shared backbone for the domain classifier and main model is efficient, adding minimal extra parameters, and these can be discarded at inference, making the model lightweight.
+ The method does not require significant architectural changes, making it relatively easy to integrate into existing DNN frameworks.
+  The authors provide substantial experimental evidence across multiple benchmarks, showcasing the effectiveness of DFP compared to state-of-the-art methods.

**Weaknesses:**

- The use of domain-specific perturbations might risk overfitting to domain characteristics present in the training set, which could limit generalizability.
- The paper seems to focus more on the empirical side and might lack a robust theoretical underpinning for why and how DFP works.
- The effectiveness of DFP assumes the availability of accurate domain labels, which might not always be feasible in real-world applications.

**Questions:**

- How does the performance of DFP change in scenarios where domain labels are noisy or inaccurate?
- Have you investigated the potential overfitting to domain-specific noise, especially when domain characteristics are subtle or complex?

---

### Official Review · Reviewer_VyGr · 2023-10-30

**Soundness:** 2 fair
**Presentation:** 3 good
**Contribution:** 2 fair
**Rating:** 5
**Confidence:** 5

**Summary:**

The paper introduces a new method for domain generalization called domain feature perturbation (DFP). This approach uses a domain classifier to create perturbations for domain-specific features, with the goal of decreasing the model's reliance on these features. The authors carried out comprehensive tests on various domain generalization datasets. The results show that DFP is effective and, in some cases, offers better out-of-domain (OOD) performance compared to leading methods.

**Strengths:**

**Originality:**
The paper presents a unique approach to domain generalization with the introduction of domain feature perturbation (DFP). This technique of using a domain classifier to perturb domain-specific features is a fresh perspective in the field.

**Quality:**
The comprehensive experiments conducted on multiple domain generalization datasets vouch for the robustness of the research. Their results, which in some cases surpassed state-of-the-art methods, further attest to the high quality of their work.

**Clarity:**
The paper is well-structured, with clear explanations of the DFP methodology and its underlying principles. The use of domain-specific perturbations and the aim of reducing model dependence on such features is lucidly presented.

**Significance:**
By addressing the challenge of domain generalization, the paper makes a contribution to the broader machine learning community.

**Weaknesses:**

**Weaknesses of the Paper:**

**1. Lack of Novelty with Domain Classifier:**
The utilization of a domain classifier in domain adaptation/generalization is not a novel concept. While the authors innovatively used the domain classifier to obtain the pre-activation gradient and perturb the pre-activation in the second forward pass, the foundational idea is not groundbreaking.

**2. Incomplete Insight on Gradient Variance:**
In Section 2.2, the authors mention using the preactivation gradient as noise variance based on its likelihood of having larger gradient magnitudes. However, more insight is desired. Specifically, the rationale behind only employing the gradient for variance and not considering perturbations in the mean remains unexplained.

**3. Overlooking Probabilistic Model Approaches:**
Considering the model's probabilistic nature, the authors could have explored sampling multiple 'n' values from the distribution to enhance model uncertainty. This is akin to techniques observed in Variational Autoencoders (VAE) or Variational inference models.

**4. Insufficient Experiments:**
The experiments presented, specifically Tables 1, 2, and 3, primarily highlight DFP's superiority over ERM. Consolidating these findings into one table in the main text or merging them and moving extras to the appendix could make space for more insightful experiments. Additionally, the absence of an experiment that omits the domain classifier in the second forward pass is a missed opportunity. Finally, it would be intriguing to see the results if an object classifier were added during the first-forward pass, and then the pre-activation gradient from that is used to perturb the second forward pass.

**Questions:**

**Questions and Suggestions for the Authors:**

**Questions:**

1. **Insight on Gradient Use:** One of the primary concerns revolves around the gradient's employment. Can the authors provide deeper insights into their choice of using the preactivation gradient for perturbation? What led to this particular decision, and how does it enhance the model's performance?

2. **Rationale for Gradient Variance:** Why was the gradient only used for variance in the perturbation and not considered for the mean? What was the underlying reasoning for this choice, and are there any potential benefits or drawbacks to this approach?

**Suggestions:**

1. **Highlight Gradient Contribution in Introduction:** Given that the use of the gradient for perturbation is a significant contribution of this paper, it would be beneficial to emphasize this aspect more prominently in the introduction. Detailing its importance upfront can set the stage for readers to understand its impact throughout the paper.

2. **Reorganization of Table 6:** Consider incorporating DFP's results directly into Table 6. Doing so would offer a clearer and more direct comparison, allowing readers to immediately discern the advantages of DFP. This presentation would make the benefits of DFP more evident and strengthen the paper's argument.

---

### Official Review · Reviewer_H66K · 2023-10-31

**Soundness:** 2 fair
**Presentation:** 3 good
**Contribution:** 2 fair
**Rating:** 3
**Confidence:** 4

**Summary:**

This work tackles multi-source domain generalization task. It aims at reducing the dependence of the prediction model on domain-specific features. It trains a domain classifier in conjunction with the main prediction model, the domain classifier is used to provide perturbation in multi hidden representations, the amplitude of the perturbation is adaptively guided by the gradient of the domain classifier. the institution to do so is that the features with high absolute gradient value (obtained from the domain classifier) are highly domain-specific and a perturbation on these features can make the model less depend on these features.

**Strengths:**

the method is clearly developed, the utilizing of the perturbation in hidden layers make sense. The authors propose to learn a selective perturbation on hidden representations. Overall, the method is easy to complete.

**Weaknesses:**

the experimental results are not very sound. Recently, there are many SOTA methods have been proposed, most of them have a performance near 88.0% on PACS with Resnet50 as backbone.
The experiments in Figure2 seems not strong enough to show the effectiveness of the proposed perturbation technique, the overall gradient similarities are smaller than 0.004, and enhancement in this scale is neglectable.

**Questions:**

1.	The definition of scaler and vector should be consistent, for example in equation (9), x y, z, and eta are vectors, e.g. use blod in lower case for vector, and use non-bold in lower case for scalar, and so on.
2.	Why the authors use different steps for the three datasets? And 5100 seems to be a very special selection. How to select a proper step number when one comes to a new dataset?
The compared methods are not so SOTA, for example, the results in Table2 is much lower than some of existing methods, what about the performance when compared to SWAD[4], DIWA[1],Fishr[2], SAGM[3]?
3.	In Figure 2, the gradient similarities of both methods under all settings are very low, smaller than 0.004, which makes the effectiveness of the proposed method not significant.
4.	Table 6 is suggested to be combined with the compared tables.


[1]Rame, Alexandre, et al. "Diverse weight averaging for out-of-distribution generalization." Advances in Neural Information Processing Systems 35 (2022): 10821-10836.
[2]Rame, Alexandre, Corentin Dancette, and Matthieu Cord. "Fishr: Invariant gradient variances for out-of-distribution generalization." International Conference on Machine Learning. PMLR, 2022.
[3]Wang, Pengfei, et al. "Sharpness-aware gradient matching for domain generalization." Proceedings of the IEEE/CVF Conference on Computer Vision and Pattern Recognition. 2023.
[4]Cha, Junbum, et al. "Swad: Domain generalization by seeking flat minima." Advances in Neural Information Processing Systems 34 (2021): 22405-22418.